# Sr[Li$_2$Al$_2$O$_2$N$_2$]:Eu$^{2+}$—A high performance red phosphor to brighten the future

Gregor J. Hoerder[1], Markus Seibald[2], Dominik Baumann [2], Thorsten Schröder[2], Simon Peschke[2], Philipp C. Schmid[2], Tobias Tyborski[2], Philipp Pust[3], Ion Stoll [3], Michael Bergler[3], Christian Patzig[4], Stephan Reißaus[4], Michael Krause[4], Lutz Berthold[4], Thomas Höche [4], Dirk Johrendt [5] & Hubert Huppertz [1]

Innovative materials for phosphor converted white light-emitting diodes are in high demand owing to the huge potential of the light-emitting diode technology to reduce energy consumption worldwide. As the primary blue diode is already highly optimized, the conversion phosphors are of crucial importance for any further improvements. We report on the discovery of the high performance red phosphor Sr[Li$_2$Al$_2$O$_2$N$_2$]:Eu$^{2+}$ meeting all requirements for a phosphor's optical properties. It combines the optimal spectral position for a red phosphor, as defined in the 2016 Research & Development-plan of the United States government, with an exceptionally small spectral full width at half maximum and excellent thermal stability. A white mid-power phosphor-converted light-emitting diode prototype utilising Sr[Li$_2$Al$_2$O$_2$N$_2$]:Eu$^{2+}$ shows an increase of 16% in luminous efficacy compared to currently available commercial high colour-rendering phosphor-converted light-emitting diodes, while retaining excellent high colour rendition. This phosphor enables a big leap in energy efficiency of white emitting phosphor-converted light-emitting-diodes.

[1] Institute of General, Inorganic and Theoretical Chemistry, University of Innsbruck, Innrain 80–82, A-6020 Innsbruck, Austria. [2] OSRAM Opto Semiconductors GmbH, Mittelstetter Weg 2, D-86830 Schwabmünchen, Germany. [3] OSRAM Opto Semiconductors GmbH, Leibnizstr. 4, D-93055 Regensburg, Germany. [4] Fraunhofer Institute for Microstructure of Materials and Systems IMWS, Walter-Hülse-Straße 1, D-06120 Halle, Germany. [5] Department Chemistry, Ludwig Maximilian University of Munich, Butenandtstr. 5-13, D-81377 Munich, Germany. Correspondence and requests for materials should be addressed to H.H. (email: Hubert.Huppertz@uibk.ac.at)

The worldwide energy demand is at an all-time high and will continue to rise further due to global population growth and ever increasing industrialisation[1]. According to the International Energy Outlook 2017[2], the worldwide energy consumption is expected to rise by 28% between 2015 and 2040, which is equivalent to $4,72 \times 10^{10}$ MWh. This increase in energy consumption leads to several challenges, the most prominent being the impact of energy production on the world climate and the dwindling reserves of fossil fuels[3]. Therefore, the development of energy-saving technologies is more important than ever.

In 2015, the lighting sector in the US was responsible for ~15% of the country's entire electricity consumption. The US department of energy has estimated the potential energy savings due to SSL (Solid State Lighting) technology to be up to 75% in the year 2035 compared to a non-SSL scenario[4]. The commercial breakthrough for SSL came with the invention of phosphor converted white LEDs[3,5–12], generally referred to as white pc-LEDs. This technology employs a primary blue LED-chip, while parts of the emitted blue light are converted to longer wavelengths by using mixtures of luminescent materials to obtain white light (additive colour mixing)[13]. This approach yields high colour quality (e.g. colour rendering index (CRI) values greater than 90) and high luminous efficacy of radiation (LER). The colour rendering index (CRI) is a dimensionless index, describing the ability of a white light source to render the colours of illuminated objects to the human visual perception[14]. For high-CRI applications an additional index, R9, is included to describe the rendering of deep red coloured objects[14]. The luminous efficacy of radiation (LER) is defined as the ratio of the total luminous flux (lumens) to the spectral power (watts)[15]. A high CRI value is currently obtained at the cost of LER, as can be seen in commercially available illumination grade pc-LEDs[16]. This trade-off is mostly caused by broad emission bands of red phosphors and the reduction in sensitivity of the human eye towards the near infrared region[14,17]. In order to sufficiently cover the red spectral region (emission maximum 610–620 nm)[18], while minimizing the efficacy loss caused by low eye sensitivity at long wavelengths, new red phosphors with exceptionally narrow emission bands near the optimal wavelength are required[14,18].

Until now, only very few red emitting phosphors fulfil the high demands of commercial LED applications; namely a high conversion efficiency at LED operating conditions as well as degradation resistance over LEDs lifetime. For example, $Eu^{2+}$ doped nitride materials like $(Ba,Sr)_2Si_5N_8:Eu^{2+}$ [13,19] ($\lambda_{max}$ ~590 to 625 nm, full width at half maximum (FWHM) ~2050 to 2600 cm$^{-1}$ or ~71 to 101 nm), $(Ca,Sr)SiAlN_3:Eu^{2+}$ [20,21] ($\lambda_{max}$ ~610 to 660 nm, FWHM ~2100 to 2500 cm$^{-1}$ or ~78 to 108 nm) and the recently published $Sr[LiAl_3N_4]:Eu^{2+}$ [22], ($\lambda_{max}$ ~654 nm, FWHM ~1180 cm$^{-1}$ or ~50 nm), commonly abbreviated as SLA, have made their way into commercially available white pc-LEDs. The first two examples both suffer from very large FWHM and therefore low LER values, which is unattractive for high-performance applications. White pc-LEDs are characterized by a correlated colour temperature (CCT), which is the chromaticity position along the Planckian locus[15]. $Sr[LiAl_3N_4]:Eu^{2+}$-containing warm-white pc-LEDs (CCT equals 2700 K, CRI greater than 90) have already demonstrated an LER gain of 14% compared to a commercial high-CRI pc-LED due to the significantly reduced FWHM of $Sr[LiAl_3N_4]:Eu^{2+}$ [22]. However, the rather long and hardly tuneable spectral position of $Sr[LiAl_3N_4]:Eu^{2+}$ still causes considerable amounts of radiation in spectral regions with low human eye sensitivity. The only known red-emitting phosphor combining a very narrow FWHM (43 nm, 1170 cm$^{-1}$) with an emission maximum in the desired spectral region ($\lambda_{max}$ equals 615 nm) is $SrMg_3SiN_4:Eu^{2+}$ [23], which suffers from high thermal quenching, rendering it unsuitable for application in pc-LEDs[24].

Several red-emitting phosphor materials have been reported in recent years, *e.g.* nitrido(alumo/magneso)silicates such as: $Li_2SiN_2:Eu^{3+}$ ($\lambda_{max}$ ~612 nm, line emission)[25], $CaAlSiN_3:Eu^{2+}$ ($\lambda_{max}$ ~630 nm, FWHM ~86 nm or 2166 cm$^{-1}$)[26,27], $Li_2Ca_2[Mg_2Si_2N_6]:Eu^{2+}$ ($\lambda_{max}$ ~638 nm, FWHM ~1513 cm$^{-1}$ or ~86 nm)[28], $Sr[Mg_3SiN_4]:Eu^{2+}$ ($\lambda_{max}$ ~615 nm, FWHM ~1170 cm$^{-1}$ or ~43 nm)[23], and $Ba[Mg_3SiN_4]:Eu^{2+}$ ($\lambda_{max}$ ~670 nm, FWHM ~1970 cm$^{-1}$ or ~88 nm)[29], or nitridolithoaluminates such as: $Sr_4[LiAl_{11}N_{14}]:Eu^{2+}$ ($\lambda_{max}$ ~670 nm, FWHM ~1880 cm$^{-1}$ or ~85 nm)[30], $Ca_{18.75}Li_{10.5}[Al_{39}N_{55}]:Eu^{2+}$ ($\lambda_{max}$ ~647 nm, FWHM ~1280 cm$^{-1}$ or ~53 nm)[31], and $Ca[LiAl_3N_4]:Eu^{2+}$ ($\lambda_{max}$ ~668 nm, FWHM ~1333 cm$^{-1}$ or ~60 nm)[32]. None of which meet the demands for a high performance red phosphor; namely a narrow band emission between 610 and 620 nm[18] combined with excellent stability of its luminescence properties[33]. Besides europium-doped compounds, there are two other important approaches to red phosphors for pc-LEDs: Quantum dots and $Mn^{4+}$ doped fluorides[34]. While quantum dots, usually containing Cd, show great potential as red phosphors, their expensive production and high toxicity are their main drawbacks[35]. $Mn^{4+}$ doped fluorides ($\lambda_{max}$ ~630 to 640 nm) on the other hand have the advantage of exhibiting a line emission, which is inherently narrow. Yet, they suffer from long decay times, which mostly limits their use to low power LEDs, and they usually require hydrofluoric acid in their syntheses, which is highly undesirable for health and safety reasons[36]. In comparison with these two compound classes, the main drawback of $Eu^{2+}$ doped phosphors are their broad emission bands. Therefore, a narrow emission band is the main criterion for which $Eu^{2+}$ doped red phosphors have to be optimized[12,33].

$Sr[Li_2Al_2O_2N_2]:Eu^{2+}$ (abbreviated SALON) combines the performance of state of the art $Eu^{2+}$ based red phosphors with an emission maximum exactly at the desired wavelength of 614 nm[18]. A prototype white pc-LED using SALON confirmed an additional LER benefit of 16% compared to a $Sr[LiAl_3N_4]:Eu^{2+}$ containing reference LED[22], while sustaining an excellent colour rendering index (CRI equals 91, R9 equals 41).

## Results

**Crystal Structure**. The crystal structure of SALON ($Sr[Li_2Al_2O_2N_2]:Eu^{2+}$) was solved and refined based on single-crystal X-ray diffraction data (for detailed information on the synthesis and structure see Supplementary Table 1 to Supplementary Table 5)[37–39]. SALON crystallizes in a hitherto unknown structure type in the tetragonal space group $P4_2/m$ (no. 84) with the unit-cell parameters $a$ equals 7.959(2) Å and $c$ equals 3.184 (1) Å. The structure is an ordered variant of the $UCr_4C_4$-structure type with Sr on the corresponding uranium site, an ordering of aluminium and lithium on the corresponding chromium site and an ordering of nitrogen and oxygen on the corresponding carbide site. Based on the data quality and the different scattering factors, the location and occupation of the Sr, Li and Al sites could be unambiguously determined. Two kinds of tetrahedra form a highly condensed network hosting the strontium cations in one of the resulting channels (Fig. 1a). In the first type of tetrahedra (T1, grey tetrahedra), aluminium is coordinated by three nitrogen and one oxygen atom forming a $[AlON_3]^{8-}$ unit. In the second type (T2, orange tetrahedra), lithium is coordinated by one nitrogen and three oxygen atoms forming a $[LiO_3N]^{8-}$ unit. Each oxygen or nitrogen atom acts as a fourfold bridging atom leading to a sphalerite-like degree of condensation $\kappa$, which is equivalent to the atomic ratio of (Li,Al)/(O,N) equals 1. This is unusual for anionic 3D tetrahedra networks (an illustration of this network can be found in Supplementary Fig. 1).

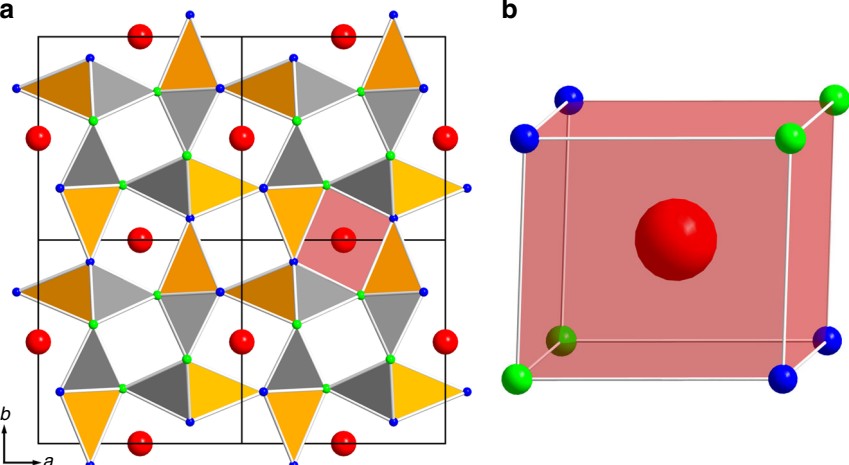

**Fig. 1** Structural overview of SALON. Red spheres represent strontium, blue spheres oxygen, and green spheres nitrogen atoms. The $[LiO_3N]^{8-}$ tetrahedra are shown in orange, the $[AlON_3]^{8-}$ tetrahedra in grey, and the red polyhedron represents the cube-like coordination of $Sr^{2+}$. **a** Viewing of a $2 \times 2 \times 2$ supercell of $Sr[Li_2Al_2O_2N_2]:Eu^{2+}$ along $[00\bar{1}]$. **b** Perspective viewing of the eight-fold coordination of $Sr^{2+}$ by O and N

The tetrahedra form a network of vierer rings[40] arranged in three types of channels along the [001] axis. These channels are connected via common tetrahedra. Two of the channels are empty and solely built up from T1- or T2-tetrahedra. The third type of channel, which hosts the strontium cations, is built up from alternating T1- and T2-tetrahedra. As the strontium cations are located at a centre of inversion, an unprecedented coordination incorporating four nitrogen (Sr–N: 2.760(5) Å) and four oxygen (Sr–O: 2.659(4) Å) atoms is obtained, resulting in a highly symmetrical cube-like coordination (Fig. 1b). The $Eu^{2+}$ activator partly replaces $Sr^{2+}$, as the ionic radii of $Eu^{2+}$ (1.39 Å) and $Sr^{2+}$ (1.40 Å) are nearly identical[41]. The structure model was confirmed by BLBS[42] and CHARDI[43] calculations (see Supplementary Table 6 and 7), which are all in good agreement with the single-crystal X-ray diffraction data.

Due to the limitations of X-ray diffraction, the ordering of the nitrogen and oxygen atoms could not be determined with absolute certainty. However, the obtained residuals and figures of merit were much better using this approach. With a ratio of O/N equals 1/1, this model is in good agreement with the elemental analysis data (see next chapter). Additionally, one has to keep in mind that the absence of the described ordering would lead to a variety of different local coordination environments for the europium cations, which would in turn cause a definitely broader emission band. To confirm the above mentioned ordering MAPLE (Madelung part of lattice energy) calculations have been performed. The results of these calculations are in good agreement with the other data and the oxygen/nitrogen-ordering is therefore a reasonable assumption. For the future, an in-depth analysis of the oxygen/nitrogen ordering via neutron diffraction could be employed.

To ensure that the crystal used for the structure determination is representative for the whole sample, a powder diffraction pattern of a bulk sample was recorded. A Rietveld refinement of the PXRD data revealed that 93 wt% of the bulk material was indeed the afore described phase $Sr[Li_2Al_2O_2N_2]$, with impurities of SrO (7 wt%) and AlN (not quantifiable). For further details see Supplementary Note 1, Supplementary Table 8, and Supplementary Fig. 2.

Although SALON was found to be an oxonitride compound with an oxygen to nitrogen ratio of 1:1, it exhibits some of the structural motives typical for nitride compounds. This is very unusual as oxygen rich compounds usually do not exhibit the

same connectivity as purely nitride based compounds. To the best of our knowledge, this is the first time that a cube-like coordination, similar to the one found in many nitride based red phosphors, could be realized in an oxonitride with an oxygen/nitrogen ratio as high as 1:1.

**Elemental analysis**. Analytical scanning transmission electron microscopy (STEM) in combination with energy-dispersive X-ray spectrometry (EDX) confirmed the composition of SALON. The STEM micrograph (Supplementary Fig. 3) shows two phosphor particles, both with diameters of a few hundred nanometres (100 nm at their thinnest, 500 nm at their thickest point), embedded in a SrO-matrix. This matrix is the result of the sample preparation process which was carried out on a larger SALON particle and not inherent to the sample. EDX element maps, obtained by plotting the lateral distribution of the EDX signal intensity of the respective X-ray lines within the spectrum, indicate that the crystals are rich in nitrogen, aluminium, and strontium, and also contain oxygen (see Fig. 2a). The surrounding matrix is rich in strontium and oxygen. A quantification of the EDX spectra acquired from the phosphor particles resulted in an element ratio of Sr:Al:O:N approximately equal to 1:1.8:2.1:1.7. Within the accuracy of the measurement, this result is very close to Sr:Al:O:N equals 1:2:2:2, thus in agreement with the X-ray diffraction data.

Since the EDX setup did not allow for the detection of lithium, time-of-flight secondary ion mass spectroscopy (ToF-SIMS) was performed to verify the lithium content of the phosphor. As Fig. 2b shows, the ToF-SIMS analysis unambiguously detected Li with homogeneous distribution in the phosphor particle. In order to gain quantitative information concerning the composition, ToF-SIMS depth profiles were acquired from a phosphor particle, and also from a reference with known composition, namely $Sr[LiAl_3N_4]$. Within these depth profiles, the signal intensities of the $Li^+$ secondary ions, divided by the signal intensities of the $Sr^+$ secondary ions, approached a linear level in the phosphor bulk below the surface. As can be seen in Fig. 2c, the intensity of the $Li^+$ signal in SALON is higher than that of the $Sr[LiAl_3N_4]$ reference by a factor of 2.4, which is close to 2. The remaining 0.4 difference can be attributed to matrix effects, which cannot be accounted for as there is no established reference available, due to the fact that SALON is the only known representative of this

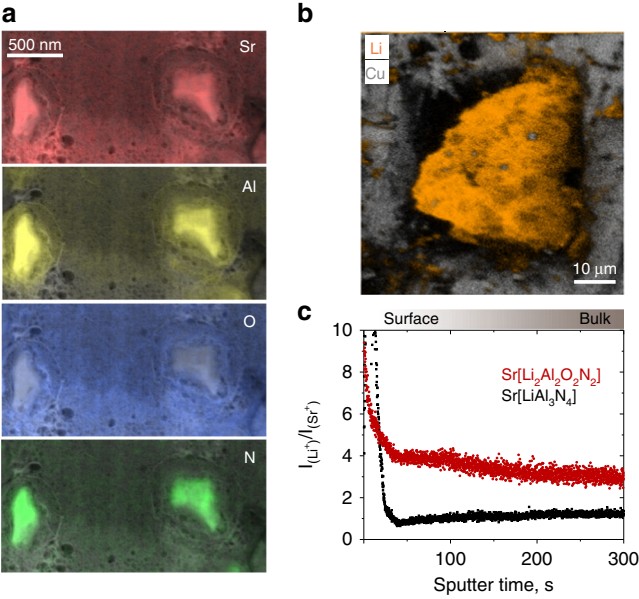

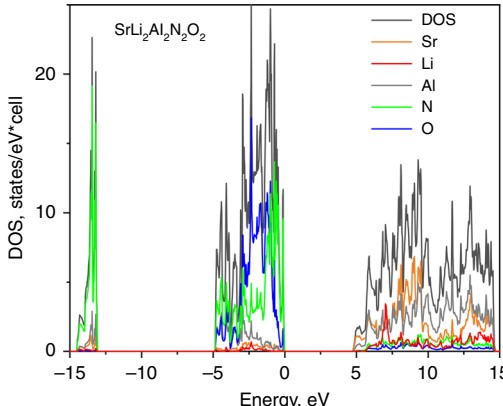

**Fig. 3** Projected total density of states (DOS) obtained from Linear combination of atomic orbitals and atom-resolved partial electronic density-of-states (pDOS) of SALON. The energy zero is taken at the Fermi level

**Fig. 2** Scanning transmission electron microscopy combined with energy-dispersive X-ray spectroscopy (STEM/EDX) and time of flight secondary ion mass spectrometry (ToF-SIMS) analyses of SALON. **a** Qualitative false-colour representation of the lateral distribution of the elements Sr, Al, O, and N. **b** Qualitative false-colour representation of the lateral distribution of the elements Li and Cu for a $Sr[Li_2Al_2O_2N_2]:Eu^{2+}$ phosphor particle that has been placed into a pocket of a copper transmission electron microscopy grid with 50 μm mesh width. **c** Time of flight mass spectrometry depth profile of the $Li^+$ secondary ions signal intensities in $Sr[LiAl_3N_4]$ and $Sr[Li_2Al_2O_2N_2]$, divided by the respective signal intensities of the $Sr^+$ secondary ions. Below the particle surface, the normalized signal intensities approach a linear level (further details are provided in the Method section)

compound class. Hence, by combination of the EDX and ToF-SIMS results, taking electro-neutrality into account, the only possible composition of SALON was determined to be $Sr[Li_2Al_2O_2N_2]$, which confirmed the results obtained from single-crystal X-ray diffraction.

**DFT Calculations**. The DFT (density functional theory) electronic band-structure calculations identify $Sr[Li_2Al_2O_2N_2]$ as an insulating material with highly ionic bonding character. $Sr[Li_2Al_2O_2N_2]$ has a calculated band gap of 4.9 to 5.3 eV (depending on the mBJ (modified Becke-Johnson) parametrisation), which is larger than the band gap of $Sr[LiAl_3N_4]$ (4.14 eV) due to the stronger ionic character in the oxonitride material[44]. The band gap has also been estimated by means of diffuse reflectance spectroscopy and analysis via the Tauc-method[45], resulting in a value of ~4.4 eV (for details see Supplementary Fig. 4). This value is well within the expected scope, as optical band gaps are usually slightly smaller than the band gaps calculated via DFT. The calculated large band gap is in good agreement with the very low thermal quenching (see next chapter). Figure 3 shows the total density-of-states (DOS) and atom-resolved partial density-of-states curves (pDOS). The latter gives the contributions of the constituent elements to the total DOS. The valence bands originate from the oxygen and nitrogen 2p-orbitals and are very narrow, which is typical for salt-like ionic compounds. Empty states of strontium and aluminium form the bottom of the conduction band, while the empty states of Li are at slightly higher energies.

Integrations of the partial atom-resolved pDOS from the projected LCAO (linear combination of atomic orbitals) base yield atom charges that reflect the expected ionic distribution as well as the charges obtained from Bader's atom-in-molecules (AIM) approach (further details are provided in the Method section and Supplementary Table 9)[46]. The calculated bulk-modulus of SALON is 107 GPa and thus comparable with that of silicon (100 GPa).

**Luminescence and LED efficacy data of SALON**. $Eu^{2+}$-doped samples of SALON exhibit intense red luminescence when excited with UV to green light. Figure 4a shows the excitation spectrum (grey curve) and the emission spectrum (red curve) of a SALON bulk sample. The emission of the single-crystal used for the structure determination has also been measured and is very similar to the emission of the powder (see Supplementary Note 2 and Supplementary Fig. 5). Therefore, the luminescence properties of the bulk sample can clearly be associated with SALON. This also rules out the possibility that SrO, which is present as a side phase in the bulk sample of SALON and can also exhibit a red emission when doped with $Eu^{2+}$, contributes to the observed emission spectra[47]. The excitation spectrum's maximum is located at roughly 450 nm, which is in the region of common wavelengths for primary blue LEDs. Additionally, the broad absorption band can be assigned to parity-allowed $4f^7(^8S_{7/2}) \rightarrow 4f^6(^7F)5d^1$ transitions within the $Eu^{2+}$ activator. SALON exhibits a narrow-band emission (FWHM equals 48 nm, $1286 cm^{-1}$, 0.1594 eV) in the red spectral region ($\lambda_{max}$ equals 614 nm, perfectly meeting the requirement defined for red conversion phosphors in the solid state lighting R&D plan 2016). In comparison to $Sr[LiAl_3N_4]:Eu^{2+}$ ($\lambda_{max}$ approximately equals 654 nm; FWHM approximately equals 50 nm, $1180 cm^{-1}$, 0.1463 eV, purple curve[22]), a very good red phosphor which has received significant attention, the main advantage of SALON lies in the position of its emission maximum, which is located at higher energies than the emission maximum of SLA and thereby significantly increases the emission's overlap with the human eye sensitivity curve (dotted black curve in Fig. 4a). The optimized spectral position of the SALON phosphor leads to an LER increase by a factor of roughly 3.5 compared to $Sr[LiAl_3N_4]:Eu^{2+}$ (266 $lm/W_{opt}$ for SALON vs. 77 $lm/W_{opt}$ for $Sr[LiAl_3N_4]:Eu^{2+}$, both at roughly 300 K)[22] regarding full-conversion red-emitting pc-LEDs built up from a primary blue LED and one single red-emitting phosphor. The relative blue shift of SALON's emission

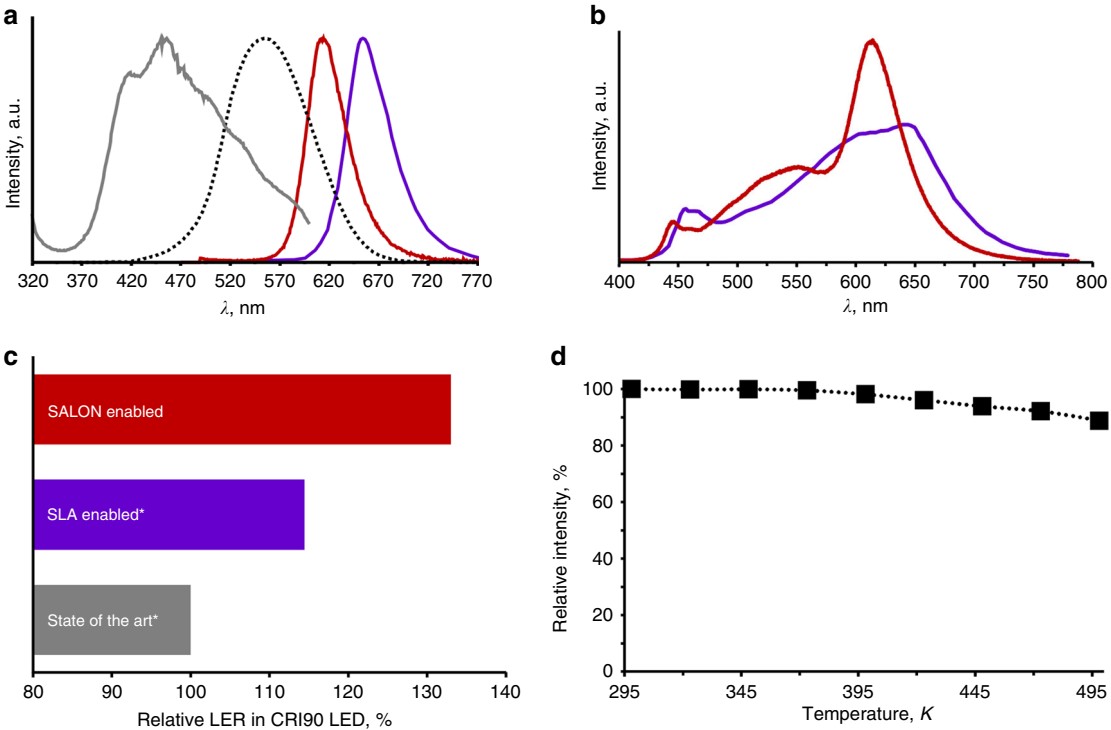

**Fig. 4** Photoluminescence properties of SALON. **a** Normalized excitation (grey, for the emission at $\lambda_{max}$ equals 614 nm) and emission spectrum (red, excited with $\lambda_{exc}$ equals 460 nm) of SALON in comparison to Sr[LiAl$_3$N$_4$]:Eu$^{2+}$ [22], reference (purple, data taken from literature) and the human-eye sensitivity curve (black dotted). **b** Comparison of two warm-white pc-LEDs (CCT equals 2700 K, CIE coordinates (SALON based LED): x/y equals 0.463/0.415; CIE coordinates (SLA based LED): x/y approximately equal 0.455/0.405, spectra normalized to integral intensity) with high colour rendering index (CRI greater than 90, R9 greater than 40). Purple curve taken from literature (Sr[LiAl$_3$N$_4$]:Eu$^{2+}$ as red phosphor), the red curve represents a mid-power pc-LED using the SALON phosphor as red component. **c** Relative luminous efficacy of radiation (LER) values for different warm-white pc-LED solutions (data normalized to the state of the art). Solutions marked with an asterisk are taken from the literature[22]. The solid red bar represents the relative LER value for the SALON containing phosphor solution with R9 greater than 40. **d** Relative photoluminescence intensity of SALON measured from 298 K to 500 K

maximum is caused by a less pronounced nephelauxetic effect and a lower crystal field splitting due to the presence of oxygen atoms in the Eu$^{2+}$-coordination sphere (Fig. 1b), whereas only nitrogen atoms are present in the case of Sr[LiAl$_3$N$_4$]:Eu$^{2+}$. In addition to the well suited spectral position of SALON for red components in high-performance white LEDs, its small FWHM of 48 nm makes a huge step on the way towards the goal of reducing the FWHM of red phosphors to strongly increase the LED's overall efficacy[18].

To ensure that SALON does not suffer from pronounced thermal quenching, the temperature dependence of the integrated photoluminescence intensity was investigated for a powder sample from 298 K to 500 K (see Fig. 4d and Supplementary Fig. 6). Low-temperature luminescence investigations showed a gradual reduction of the FWHM yielding 41 nm at 15 K. Interestingly, the FWHM reduction mainly occurs on the high energy side of the emission spectrum (see Supplementary Fig. 7). Even at typical LED operating temperatures of 420 K, the integrated light output of SALON drops by only 4% compared to the initial intensity at room temperature (298 K), while compounds, which suffer from considerable thermal quenching like SrMg$_3$SiN$_4$:Eu$^{2+}$ [23], have already lost over 90% of their maximum intensity at this temperature (420 K)[48]. SALONs outstanding thermal quenching behaviour surpasses the performance of most other phosphors (e.g. CaAlSiN$_3$:Eu$^{2+}$, Ba[Li$_2$(Al$_2$Si$_2$)N$_6$]:Eu$^{2+}$)[26,49] and is a valuable quality for application in pc-LEDs. A simultaneous shift in the CIE (Commission Internationale de l'Éclairage) colour coordinates, x/y equals 0.654/0.346 to x/y equals 0.625/0.374 ($\Delta x/\Delta y$ equals 0.029/0.028), and LER-values of SALON was also recorded in the range from 298 to 473 K. This is a rather small colour shift and comparable to the shift found in Sr[LiAl$_3$N$_4$]:Eu$^{2+}$, from x/y equals 0.693/0.306 to x/y equals 0.668/0.330 ($\Delta x/\Delta y$ equals 0.025/0.024) between 303 K and 465 K. Finally, the quantum efficiency (number of converted photons emitted/number of photons absorbed in %) of SALON was determined to be greater than or equal to 80%. This value should be comparable to the literature value for the "internal quantum efficiency" of Sr[LiAl$_3$N$_4$]:Eu$^{2+}$ of 76%[22].

The luminescence decay time of SALON has been determined by a Time-Correlated Single Photon Counting (TCSPC) experiment (see Supplementary Fig. 8). The relaxation constant $\tau$ of 790 ns is in good agreement to those determined for other red-emitting Eu$^{2+}$-doped phosphors[34,50].

Although SALON powder in this early state without any protective coating is not stable against direct exposure to water, it was possible to construct a prototype pc-LED, where SALON was shown to be stable enough to be utilized in silicone under ambient conditions up to at least 250 °C. As demonstrated for SLA, even highly sensitive nitride materials can be stabilized enough to be used in commercial pc-LEDs[51]. A similar process will be necessary for this material in the future.

The spectrum of this warm-white prototype pc-LED containing the SALON material blended in a two phosphor mix in combination with a green Lu$_3$(Al/Ga)$_5$O$_{12}$:Ce$^{3+}$ [52], phosphor

(Fig. 4b, red curve) was compared to a pc-LED spectrum taken from literature, with a three phosphor mix (Fig. 4b, purple curve; $Lu_3Al_5O_{12}$:$Ce^{3+}$ [53], (green), $(Ba,Sr)_2Si_5N_8$:$Eu^{2+}$ [13,19] (amber) and $Sr[LiAl_3N_4]$:$Eu^{2+}$ [22], (red)). Both pc-LEDs were steered to a correlated colour temperature (CCT) equals 2700 K (CIE coordinates (SALON based LED): x/y equals 0.463/0.415; CIE coordinates (SLA based LED): x/y approximately equal 0.455/0.405). The SALON pc-LED achieved a remarkable benefit of $+$ 16% regarding the spectral LER in comparison to the $Sr[LiAl_3N_4]$:$Eu^{2+}$ containing reference or $+$ 33% to the state of the art[22] (see Fig. 4c), while sustaining an excellent colour rendering index (CRI equals 91, R9 equals 41).

## Discussion

SALON is a red-emitting phosphor with a formerly unknown host structure. It is the only known oxonitridolithoaluminate, exhibits a high degree of condensation, and shows a unique, highly symmetrical $Sr^{2+}$ coordination, incorporating distinct nitrogen and oxygen sites. SALON exhibits an exceptionally narrow-band red emission with $\lambda_{max}$ approximately equal 614 nm and a FWHM of 48 nm, already fulfilling the future standard industrial requirements for a phosphor material in high energy efficiency white pc-LED applications. The most probable reason for the narrow emission band of SALON is the high local symmetry around the $Eu^{2+}$ activator position in the crystal structure as the ordered arrangement of four oxygen and four nitrogen atoms is forming a cube-like coordination polyhedron. This highly symmetrical coordination is expected to be the reason for the narrow emission as a result of a more isotropic structural relaxation of the $Eu^{2+}$ in its excited state, which reduces the number of energetically different states involved in the emission process. All of SALON's intrinsic characteristics, which are relevant for application, namely a quantum efficiency of greater than 80%, low thermal quenching of only 4% at operating temperatures, reasonable thermal and chemical stability as well as the above mentioned favourable emission properties, mark it as a highly promising compound which will, once thoroughly optimized, make for a formidable red phosphor in future white pc-LEDs. A prototype mid-power white LED using the SALON phosphor as its red component already demonstrated a significant increase in LER, while maintaining an excellent quality of light.

## Methods

**Synthesis**. Due to the moisture sensitivity of the starting materials, all manipulations were carried out using an inert gas filled glovebox (MBraun, Garching, Germany; $O_2$ less than 1 ppm, $H_2O$ less than 1 ppm).

Single-crystals of $Sr[Li_2Al_2O_2N_2]$:$Eu^{2+}$ were synthesized starting from $Sr_3Al_2O_6$ (97.34 mg, 0.2358 mmol, synthesized according to the method of Garcés et al.[54].) and $LiN_3$ (23.09 mg, 0.4716 mmol, synthesized according to the method of Fair et al.[55]) as starting materials, as well as $Eu_2O_3$ (0.83 mg, 0.0024 mmol, Smart Elements, 99.99%) as doping agent, resulting in a formal europium content of 0.7%. An excess of elemental lithium (16.37 mg, 2.358 mmol, Sigma Aldrich, 99%) was added to act as a fluxing agent.

The starting materials were mixed in an agate mortar and put into a tantalum ampule. The ampule was sealed utilizing tungsten inert gas welding. During the welding process, the ampule was water-cooled to avoid untimely decomposition of the lithium azide. The closed ampule was then placed in a silica tube, which was evacuated to prevent the oxidization of the tantalum ampule. This silica tube was placed into a tube furnace and heated to 800 °C. The sample was maintained at this temperature for 100 h and subsequently cooled down to 500 °C with 0.1 °C per minute. Upon reaching 500 °C, the furnace was turned off and the sample was left to cool down to room temperature.

$Sr[Li_2Al_2O_2N_2]$:$Eu^{2+}$ was obtained as small red crystals, which exhibited an intense red luminescence upon excitation with UV to green-light.

Bulk material of $Sr[Li_2Al_2O_2N_2]$:$Eu^{2+}$, which was also used for STEM/EDX/ToF-SIMS-analysis, was synthesized by a solid-state reaction of $Sr_3N_2$ (4.438 g, 15.26 mmol, Materion), AlN (1.261 g, 30.83 mmol, Tokuyama), $Al_2O_3$ (3.143 g, 30.83 mmol, Sinochem Hebei type C), $Li_3N$ (1.074 g, 30.83 mmol, Materion), and $Eu_2O_3$ (0.081 g, 0.23 mmol, OSRAM). The starting materials were intimately mixed and placed inside of a Ni crucible. They were then heated in a tube furnace to 800 °C in a stream of 7.5% $H_2$ / $N_2$ and held at that temperature for 144 h. After the synthesis, the product was slightly inhomogeneous exhibiting small amounts of the side phases SrO and AlN. As SrO is not stable already at ambient conditions, it decomposes with time.

The rather long sintering times in the aforementioned synthesis routes result from the fact that SALON decomposes at temperatures only slightly higher than the temperature needed for its synthesis. The synthesis of a phase pure product is therefore always a subtle balance between the speed at which SALON is formed and the starting decomposition of the already formed product. The two synthesis routes described above have been optimized for the growth of single-crystals and the luminescence properties of the product, respectively.

**Crystal Structure Determination**. The intensity data of a single-crystal (approx. $10 \times 7 \times 5$ μm) were collected at room temperature on a D8 Quest Kappa X-ray diffractometer equipped with a Photon II detector (Bruker, USA). Monochromatic Cu-$K_\alpha$ radiation (λ equals 154.178 pm) was generated by a microfocus X-ray tube (Incoatec, Germany) combined with a multilayer optic. The software tools SAINT[56] and SADABS[38] were applied for data processing and multi-scan absorption correction, respectively.

The structure was solved with SHELXS[37] (version 2013/1) and refined with SHELXL[37] (version 2014/7) using WINGX[57] (version 2013/3). The full-matrix least squares refinement against $F^2$ yielded R1 equals 0.0336, wR2 equals 0.0806, and the GOF equals 1.189. The software tools PLATON[58] (version 120716) and STRUCTURE TIDY[59] were used to validate and standardise the determined crystal structure. Detailed information on refinement parameters and crystallographic data including positional parameters, anisotropic displacement parameters, selected interatomic distances, and bond angles are listed in Supplemenary Tables 1–5.

**STEM and elemental analysis**. Analytical scanning transmission electron microscopy (STEM) analysis was done using a $c_s$-abberation corrected TITAN[3] 80–300 electron microscope (FEI company, Hillsboro, USA) at 300 kV acceleration voltage. A high-angle annular dark field detector (Fischione Model 3000, Fischione company, Export, USA) and a camera length of 115 mm were used to capture high-angle annular dark field (HAADF) scanning TEM micrographs. All further analysis was carried out in a selected area of the sample (see Supplementary Fig. 3 top). The matrix, in which the particles are embedded, consists of SrO originating from the partial decomposition of the samples during the preparation. This effect is inevitable since the composition of SALON is not indefinitely stable versus the preparation with 30 keV (initial thinning)/5 keV (fine polishing) $Ga^+$ ion beams, and also versus the subsequent irradiation with 300 keV electrons in the transmission electron microscope. This is typical for materials that contain light elements such as oxygen, nitrogen, and especially lithium[60]. STEM was combined with energy-dispersive X-ray analysis (EDX), using a Super-X EDX detector that is equipped with four SDD detectors (FEI company, Hillsboro, USA), to gather lateral element distribution maps of chosen elements. Quantification of the ratio of the elements N, Sr, Al, and O was performed with a Cliff-Lorimer approach under the assumption of a sample thickness of 100 nm, using the commercially available software Esprit, v1.9 (Bruker corporation, Billerica, USA). Element mappings of the elements N, Sr, Al, and O were derived by evaluating the lateral distribution of the peak intensity, i.e., the area underlying the K edges (N, O, Al), or L edge (Sr), respectively, with an automatic routine provided by the software.

In order to avoid artefacts during the subsequent STEM-EDX analysis, the particle under investigation was fixed on a laser-cutted supporting structure (H-shaped Ge bar), prepared using a microPREP-laser cutter (3D Micromac AG, Chemnitz, Germany), while being mounted to a Cu half-ring structure. The FIB-based STEM/TEM sample preparation was done with an Auriga 40 crossbeam FIB (Carl Zeiss AG, Oberkochen, Germany), using a $Ga^+$ beam of 30 keV ion energy for initial thinning followed by a 5 keV polishing step of the lamella until electron transparency at low electron acceleration voltage was achieved. The residual thickness of the sample was identified to be ~100 nm.

Time-of-flight secondary ion mass spectrometry (ToF-SIMS) was done with a TOF-SIMS5−100 instrument (Iontof GmbH, Münster, Germany), using $Bi_1^+$ at 25 keV as primary ion species for both $Sr[Li_2Al_2N_2O_2]$ and $SrLiAl_3N_4$ particles. The high-current bunched mode and $O_2^+$ at 2 keV as sputter ion species were used for the depth profile analyses. For the element-specific high resolution imaging of the $Sr[Li_2Al_2N_2O_2]$ particle, a field of view of 60 μm $\times$ 60 μm and a raster size of 256 $\times$ 256 pixels was used in burst alignment (unbunched) mode. For the latter analysis, a preselected $Sr[Li_2Al_2N_2O_2]$ particle was positioned in the pocket of a copper TEM grid with 50 μm mesh width. To ensure that possible surface effects (contamination, surface topography etc.) do not influence the result, the ToF-SIMS depth profile analysis was not stopped until the intensity ratio I($Li^+$)/I($Sr^+$) reached a steady state within the particle bulk. The total ablation during sputtering was estimated to be some hundred nanometres. Since, due to the small size of the particles, the whole particle was thinned during the process, the exact total sputter depth could not be measured.

Concerning the lithium quantification, it is important to note that ion intensity data cannot easily be converted into elemental concentrations, as the ionization probability is not solely sensitive to the elements in question, yet also depending on the composition of the sample. This is the so-called matrix effect[61], which describes that the composition of the sample matrix itself affects the ionization probability of the elements. Nonetheless, the secondary ion intensity of a chosen element (in this

case, lithium) can be used to describe the concentration within the sample, if the matrix effect is accounted for. As SALON is the only known oxonitridolitho aluminate, there is no standard, which is similar enough to fully compensate these matrix effects. Hence, $Sr[LiAl_3N_4]$ was used as the standard for the measurements in this manuscript, as it is similar to SALON in that it is at least a nitridolithoaluminate. However, using $Sr[LiAl_3N_4]$ as a standard some matrix effects are still to be expected, which increases the error margins of the quantification.

**Spectral Characterisation**. The excitation spectrum was collected on a Fluoromax 4 spectrophotometer (HORIBA, Japan) on a compacted powder pellet of the sample in the range from 320 to 600 nm with a 1 nm step size. The emission was monitored at 614 nm.

The title compound's single-crystal emission signal was measured by exciting a single-crystal with a 460 nm laser (model Sapphire 460/10, 10 mW; COHERENT, USA). The converted light was collected using a multi-mode optical fiber (QP 600–2-VIS/BX; Ocean Optics, USA) and finally detected in a spectrometer (QE 65000; Ocean Optics, USA).

To investigate the thermal quenching behaviour, the relative photoluminescence intensities of SALON were measured from 298 to 500 K in a Fluoromax 4 spectrophotometer (HORIBA, Japan) with an excitation wavelength of 460 nm. For this purpose, a thin powder layer of SALON was prepared on top of a heating plate equipped with a thermocouple. Temperature dependent emission spectra were collected with a 25 K step size between measurements. Before each measurement, care was taken to ensure thermal equilibrium within the sample.

The quantum efficiency (number of converted photons/number of absorbed photons in %) was measured using a QUANTAURUS-QY spectrometer (Hamamatsu, Japan) equipped with a full integrating sphere (diameter approx. 8.4 cm) and a 150 W xenon excitation light source with excitation at 460 nm. As there are slightly different definitions for internal and external quantum efficiency, it is difficult to directly compare these values for different materials. However, to the best of our knowledge the here presented value is comparable to the "internal quantum efficiency" given for $Sr[LiAl_3N_4]:Eu^{2+}$ [22].

Emission spectra were also collected for different excitation wavelengths (250 nm, 400 nm and 460 nm; see Supplementary Fig. 9).

**pc-LED Prototype**. Prototype LEDs steered to the desired colour point and colour temperature were built using a standard 3030 mid-power LED package ($3 \times 3$ mm external dimension, $\lambda_{dom}$ blue LED equals 449 nm, dimension 0.89 mm × 0.56 mm ca. 0.5 mm²) with the phosphors suspended in a commercially available high refracting index silicone. Measurements were carried out with an in-house-built system based on an integrating sphere (ISP150) coupled to an Instrument Systems Compact Array Spectrometer (CAS-140B) at an operating current of 65 mA.

**DFT Calculations**. Electronic structure calculations were performed using the Vienna *ab initio* simulation package (VASP)[62,63], which is based on density functional theory (DFT) and plane wave basis sets. Projector-augmented waves (PAW)[64] were used and contributions of correlation and exchange were treated in the generalized-gradient approximation (GGA) as described by Perdew, Burke, and Ernzerhof[65]. An energy cutoff at 500 eV and dense $k$-point samplings ensured well-converged structures and smooth density-of-states. Chemical bonding was analysed using the Crystal Orbital Hamilton Population (COHP) method[66] after projecting of the PAW eigenstates basis onto localized LCAO (linear combination of atomic orbitals) crystal orbitals using the program LOBSTER[67,68]. The Bader analysis[46] implemented by Henkelman et al.[69] was used to extract charges from the electron density distribution.

Since DFT calculations based on standard LDA (local-density approximation) or GGA (generalized gradient approximation) functionals strongly underestimate the size of the electronic band gap, we used the modified Becke-Johnson exchange potential (mBJ)[70], which yields gaps with an accuracy similar to hybrid functional or GW methods.

Integrations of the partial atom-resolved pDOS from this LCAO base set yield atom charges that correctly sum up to zero. Additionally, we calculated the atom charges using the Bader atom-in-molecules (AIM) approach[46]. The results are listed below in Supplementary Table 9. The differences between the values obtained by the pDOS and Bader approach arise from the respective methods for charge calculation. The pDOS approach sums the occupied electronic states at specified atoms up to the Fermi level while the Bader charges result from integrations of the electronic charge density inside the atomic basins, which are bound by the zero flux surfaces of the charge density. The deviations of these values are well within the expected scope.

The bulk module of $Sr[Li_2Al_2O_2N_2]$ was calculated by fitting the total energy versus cell volume ($E_{total}(V)$) curves to the Birch-Murnaghan equation of state.

## Data availability

The authors declare that data supporting the findings of this study are available within the paper and its supplementary information file. The crystallographic information file may be obtained from the FIZ Karlsruhe, 76344 Eggenstein-Leopoldshafen, Germany

(fax: (+49)7247-808-666; e-mail: crysdata@fiz-karlsruhe.de, on quoting the deposition number 434640.

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

## Acknowledgements

We would like to thank Dr. Frank Jermann (OSRAM Opto Semiconductors GmbH, Schwabmünchen) for supervising the cooperation between the OSRAM GmbH and the University of Innsbruck and for the discussion of the results. We thank Dr. Stefan Lange (OSRAM Opto Semiconductors GmbH, Schwabmünchen) for the LED-spectra simulation, manuscript revisions, and result discussion as well as Dr. Franziska Hummel and Stephanie Dirksmeyer (both OSRAM Opto Semiconductors GmbH, Schwabmünchen) for the single-crystal photoluminescence measurements. Furthermore, we thank Dr. Marcus Adam (OSRAM Opto Semiconductors GmbH, Regensburg) for the low-temperature investigations and sample preparation, Andrea Böbenroth (Fraunhofer Institute for Microstructure of Materials and Systems IMWS, Halle) for the sample preparation for STEM.

## Author contributions

G.J.H. conducted the first synthesis of the title compound, optimized the synthesis for single-crystal growth and was responsible for the manuscript. M.S. was responsible for luminescence investigations and the selection of particles for STEM and ToF-SIMS experiments as well as the revision of the manuscript. D.B. conducted the solid-state synthesis, was responsible for the selection of single-crystals and crystal structure refinement as well as for the selection of particles for STEM and ToF-SIMS experiments and manuscript revision. T.S. selected the single-crystals, conducted the X-ray diffraction measurement, and performed the crystal-structure solution and refinement. S.P. optimized the solid-state synthesis and ran the thermal-quenching experiments. P.S. was responsible for the optimization of the solid-state synthesis. T.T. conducted the lifetime measurements. P.P. and I.S. constructed the LED prototype and conducted the low-temperature luminescence investigations. M.B. was responsible for the low-temperature luminescence investigations, related sample preparations, and discussion of the results. C.P. conducted parts of the STEM as well as the EDX analysis, was responsible for related discussions, and revised the manuscript. S.R. conducted the ToF-SIMS measurements. M.K. conducted the STEM sample preparation. L.B. conducted parts of the STEM analysis. T.H. guided and supervised all elemental analysis related experiments. D.J. conducted DFT calculations, was responsible for related discussions, and revised the manuscript. H.H. supervised the work conducted at the University of Innsbruck and revised the manuscript.

## Additional information

**Competing interests:** The authors declare the existence of a financial competing interest as the co-authors M.S., D.B., T.S., S.P., P.S., P.P., I.S., M.B. are employed by Osram Opto Semiconductors GmbH and parts of the research results presented in this article are part

of the following patent application: WO2018/087304A1; Osram GmbH; Inventors: M.S., D.B., T.S., S.L., G.H., G.M.A., H.H., S.P., A.M., P.S., F.H., S.D.; patent pending; covered aspects: composition, crystal structure, optical properties of $SrLi_2Al_2O_2N_2:Eu^{2+}$. The remaining authors declare no competing interests.

