## [Peer Review File · Nature Communications]

Editorial Note: This manuscript has been previously reviewed at another journal that is not operating a transparent peer review scheme. This document only contains reviewer comments and rebuttal letters for versions considered at Nature Communications. Mentions of the other journal have been redacted.

Reviewers' comments:

Reviewer #1 (Remarks to the Author):

The authors have modified their manuscript by adding most of the data requested during the previous reviewing round(s) when the paper was submitted to [redacted]. The role of the SrO impurity has been added, as well as lifetime values, a clarification on the quantum yield and the impact of color shifts. Some of the other issues have been commented on in detail in the rebuttal letter. In my opinion, the paper can in principle be accepted as it offers a valuable contribution to the field of phosphor research, provided that the following issues are adequately handled:

1) In the present version it turns out that the chemical stability of this phosphor is a serious issue. This is now indeed mentioned in the manuscript "Although SALON in this early state is not stable against direct exposure to water", and the standard mitigation route of phosphor coatings is mentioned. In this respect, it is difficult to maintain the statement in the abstract "We report on the discovery of the new high performance red phosphor $\text{Sr}[\text{Li}_2\text{Al}_2\text{O}_2\text{N}_2]:\text{Eu}^{2+}$ meeting all industry requirements for a phosphor's intrinsic properties". The chemical stability of a host is an intrinsic property of a phosphor, especially in the framework of the industry requirements. The authors also refer to reference 34 (ECS J. Solid State Sci. 448 Technol. 7, R3111-R3133 (2017) – for which the year of publication should be 2018 by the way), to state that other (nitride) phosphors do not meet the phosphor requirements, by referring to reference 34. However, in this work, chemical stability is considered as one of the six evaluation parameters ("Fifth, the inherent stability against moisture and continuous irradiation ensures the durability and longevity of the device.").

2) The authors evaded my request to add the phosphor weights for the prototype white LED. The basic information on the production of the prototype (type of blue LED, current, binder material, green phosphor, phosphor weight etc) should be added to the SI, in order for other researchers to reproduce the results described in the paper. If it turns out that the LED spectra shown in the Figure 4b are simulations, based on input of the individual LED and phosphor spectra – which would not be so surprising given the perfect match on the long wavelength side between individual spectra and the LED spectrum, despite having different geometry and detection equipment - then this should also explicitly be mentioned in the manuscript, so that it is not suggested anymore that a real pc-LED device was made. I hope this is not the case.

3) Fig. S8 states that one of the emission spectra was obtained upon excitation at 600nm. Is this correct?

Reviewer #3 (Remarks to the Author):

1. In this manuscript (MS), the title emphasizes the high performance of red phosphor for application of w-LED. It is a question what is the "high performance"? If it is only good thermal quench behavior combined with the narrow band of red emission, it is not enough to say "A high performance red phosphor to brighten the future". It is necessary to provide the comparative data which include the efficiency (? lm/W), thermal and chemical stability, humidity resistance, anti-temperature-oscillation, decay behavior and the weight of the used red phosphor powder, of the

white-light LED sources packaged by both of the best commercial red phosphor and the self-fabricated SrLi₂Al₂O₂N₂:Eu²⁺ powders, respectively, mixed with green and yellow phosphor powders, tested under the same T_c, CRI, and (x,y) coordinate. The number of the tested SrLi₂Al₂O₂N₂:Eu²⁺ samples must be at least more than five.

2.As for the "Novelty" in this MS, the new oxynitride SrLi₂Al₂O₂N₂:Eu²⁺ is developed from the well-known nitride narrow-band phosphors Sr[LiAl₃N₄]:Eu²⁺ by incorporating oxygen in it and the crystal structure is solved by X-ray single crystal refinement. So, 1) the narrow band of red emission is not one of the novelties in this MS. 2) Although the good thermal stability is demonstrated by Fig.4(d), its chemical stability is bad due to its hydrolytic property. So the stability is also not one of the novelties in this MS. 3) The new material SrLi₂Al₂O₂N₂:Eu²⁺ comes from its mother system of Sr[LiAl₃N₄]:Eu²⁺ by incorporating oxygen, so the material system is not thorough one of the novelties in this MS. The author said that the structure of the ordered O-N is the key novelty. I agree with it. But the author have to demonstrate its "high performance" in advance, which make it sense for the new structure.

3.How to transform from the space group of P1 (no. 2) for the matrix materials of Sr[LiAl₃N₄] system [P1 (no. 2) with a = 5.86631(12)Å, b = 7.51099(15)Å, c = 9.96545(17)Å , refer to Philipp Pust et al., Nature Materials 22 (2014) 1-6,] to the space group of P4₂/m (no. 84) for SrLi₂Al₂O₂N₂ by incorporation of oxygen? When does it start to transform to be the ordered O-N lattice at the critical point of the oxygen concentration? The XRD patterns of different doping concentration of oxygen should be appended. In this figures, the transitional process from disordered O-N to the order O-N lattice should be illustrated. Abrupt change or gradient change? In addition, how about the evolution of the SrO impurity with increasing oxygen concentration?

4.Because the impurity of SrO is easily decomposed in ambient, why do not remove the impurity of SrO in advance so as to receive the more confidential raw data, such as the XRD, PL and EDS, of the pure SrLi₂Al₂O₂N₂ ?

5.It is more believable to detect whether or not the superstructure exists in SrLi₂Al₂O₂N₂ by HRTEM and neutron diffraction pattern because of the low resolution of XRD for the trace phase; furthermore, it is more reliable to refine the fine structure of the ordered O-N in SrLi₂Al₂O₂N₂ by using neutron diffraction.

6.As an auxiliary means, the XPS and ESP can also be used for analysis of different coordination of O and N, and the proportion of O/N in SrLi₂Al₂O₂N₂ can be detected by Oxygen and Nitrogen Analyzer.

7.Concentration quenching of active ion is the basic phosphor performance, I feel it should be measured and presented.

Prof. Yongge Cao
Department of Physics
Renmin University of China
P.R.China

Dear reviewers,

in the following, we copied the reviewer comments into this document followed by our answers (in blue) indicating also the changes, which were made to the manuscript. Most points were revised in accordance to the suggestions. Furthermore, we clarified the other points, so we think that the manuscript fulfils the necessary criteria to be accepted. We would like to thank the reviewers for their help to improve the quality of our manuscript!

Best regards,

Hubert Huppertz

Reviewers' comments:

Reviewer #1 (Remarks to the Author):

The authors have modified their manuscript by adding most of the data requested during the previous reviewing round(s) when the paper was submitted to [redacted]. The role of the SrO impurity has been added, as well as lifetime values, a clarification on the quantum yield and the impact of color shifts. Some of the other issues have been commented on in detail in the rebuttal letter. In my opinion, the paper can in principle be accepted as it offers a valuable contribution to the field of phosphor research, provided that the following issues are adequately handled:

1) In the present version it turns out that the chemical stability of this phosphor is a serious issue. This is now indeed mentioned in the manuscript "Although SALON in this early state is not stable against direct exposure to water", and the standard mitigation route of phosphor coatings is mentioned.

In this respect, it is difficult to maintain the statement in the abstract "We report on the discovery of the new high performance red phosphor Sr[Li₂Al₂O₂N₂]:Eu²⁺ meeting all industry requirements for a phosphor's intrinsic properties". The chemical stability of a host is an intrinsic property of a phosphor, especially in the framework of the industry requirements. The authors also refer to reference 34 (ECS J. Solid State Sci. 448 Technol. 7, R3111-R3133 (2017) – for which the year of publication should be 2018 by the way), to state that other (nitride) phosphors do not meet the phosphor requirements, by referring to reference 34. However, in this work, chemical stability is considered as one of the six evaluation parameters ("Fifth, the inherent stability against moisture and continuous irradiation ensures the durability and longevity of the device.").

Answer: Of course, the chemical stability is an intrinsic property and greatly affects the industrial application of a phosphor. However, in contrast to the other properties like absorption, excitation, quantum efficiency, and thermal quenching (Requirements one to four in ECS J. Solid State Sci. 448

Technol. 7, R3111-R3133 (2018) – for which we corrected the year of publication, thank you for pointing this out) the chemical stability is not directly linked to the phosphors function and can therefore be overcome by technical means (e.g. coating). This has been done for SLA, as it also displays water sensitivity without further treatments. However, such processes have been demonstrated and described in later articles. In our case, an additional coating investigation is far beyond the scope of this first report.

Nevertheless, we agree with the reviewer that the statement “... meeting all industry requirements for a phosphor’s intrinsic properties” is phrased in a way that may suggest a higher stability than SALON actually exhibits. Therefore, we changed the statement to refer solely to optical properties and eliminated the industry reference in order to avoid the impression of reporting on a market ready phosphor.

2) The authors evaded my request to add the phosphor weights for the prototype white LED. The basic information on the production of the prototype (type of blue LED, current, binder material, green phosphor, phosphor weight etc) should be added to the SI, in order for other researchers to reproduce the results described in the paper. If it turns out that the LED spectra shown in the Figure 4b are simulations, based on input of the individual LED and phosphor spectra – which would not be so surprising given the perfect match on the long wavelength side between individual spectra and the LED spectrum, despite having different geometry and detection equipment - then this should also explicitly be mentioned in the manuscript, so that it is not suggested anymore that a real pc-LED device was made. I hope this is not the case.

Answer: The description of the prototype has now expanded to include the dominant wavelength of the primary LED, the applied current and the binder material (Supplementary Information, page 15 lines 240-245). $\text{Lu}_3(\text{Al}/\text{Ga})_5\text{O}_{12}:\text{Ce}^{3+}$ is used as the green phosphor for the SALON-LED, as mentioned in the manuscript (page 11, line 294).

As we stated in our last rebuttal letter the composition of the phosphor slurry was varied until the desired colour temperature and colourpoint (x,y) was reached. Unfortunately, the quota of phosphors added during this iterative process have not been weighed, which leaves us unable to provide phosphor weights for the prototype. However, as the used phosphors and the type of blue LED are now specified in the manuscript this approach is reproducible for any other researcher as the colourpoint and the used phosphors determine the phosphors weights. Furthermore, we want to point out, that the Nature Materials publication regarding SLA (Pust, P. *et al.* Narrow-band red-emitting $\text{Sr}[\text{LiAl}_3\text{N}_4]:\text{Eu}^{2+}$ as a next-generation LED-phosphor material. *Nat. Mater.* **13**, 891-896 (2014).) does not include any such information. As the data for the SLA containing prototype were taken from this publication, we cannot elaborate on the SLA containing prototype.

We can assure you that the emission spectrum displayed in Figure 4b is definitely not a mere simulation but a measured spectrum originating from a prototype constructed by the OSRAM GmbH. This is also unambiguously stated in the manuscript. The perfect match of the long wavelength side with the emission of SALON is the result of the fact that the SALON-LED does not need a supplementary amber phosphor and the emission intensity of $\text{Lu}_3(\text{Al}/\text{Ga})_5\text{O}_{12}:\text{Ce}^{3+}$ is practically zero in the red spectral range.

3) Fig. S8 states that one of the emission spectra was obtained upon excitation at 600nm. Is this correct?

Answer: Thank you for pointing this out to us. This statement was a mistake on our part, the spectrum in question was measured at an excitation wavelength of 460 nm and not at 600 nm. The Figure has been corrected accordingly.

Reviewer #3 (Remarks to the Author):

1. In this manuscript (MS), the title emphasizes the high performance of red phosphor for application of w-LED. It is a question what is the "high performance"? If it is only good thermal quench behavior combined with the narrow band of red emission, it is not enough to say "A high performance red phosphor to brighten the future".

It is necessary to provide the comparative data which include the efficiency (? lm/W), thermal and chemical stability, humidity resistance, anti-temperature-oscillation, decay behaviour and the weight of the used red phosphor powder, of the white-light LED sources packaged by both of the best commercial red phosphor and the self-fabricated $\text{SrLi}_2\text{Al}_2\text{O}_2\text{N}_2:\text{Eu}^{2+}$ powders, respectively, mixed with green and yellow phosphor powders, tested under the same Tc, CRI, and (x,y) coordinate. The number of the tested $\text{SrLi}_2\text{Al}_2\text{O}_2\text{N}_2:\text{Eu}^{2+}$ samples must be at least more than five.

Answer: The title, as well as the high performance statement, derive their legitimacy from the fact that SALON fulfils the requirements officially stated for future high performance LEDs by the US Department of Energy regarding the spectral position and fwhm (see US Department of Energy. Solid state lighting research and development plan. (2016); https://energy.gov/sites/prod/files/2016/06/f32/ssl_rd-plan_%20jun2016_2.pdf). Efficiency (lm/W), thermal and chemical stability, humidity resistance, and decay behaviour are all discussed and compared in the manuscript. Additionally, the description of the prototype has now expanded to include the dominant wavelength of the primary LED, the applied current and the binder material (Supplementary Information, page 15 lines 240-245).

Unfortunately, the quota of phosphors added during this iterative process have not been weighed, which leaves us unable to provide phosphor weights for the prototype. However, as the used phosphors and the type of blue LED are now specified in the manuscript, this approach is reproducible

for any other researcher as the colourpoint and the used phosphors determine the phosphors weights/ratio. We are also unable to include them for the SLA containing package as the corresponding data were taken from literature (Pust, P. *et al.* Narrow-band red-emitting Sr[LiAl₃N₄]:Eu²⁺ as a next-generation LED-phosphor material. *Nat. Mater.* **13**, 891-896 (2014).) and also that publication does not contain any details on the prototype.

Regarding the number of samples, we can assure you that the number of samples is greater than five. We have measured roughly 50 powder samples of SALON with X-ray diffraction and optical characterisation methods (most of them synthesis products from the synthesis optimisation process), and collected emission spectra of numerous isolated particles from these powders. Cell indexing was also carried out for multiple crystallites resulting in the cell obtained from the structure determination.

2. As for the "Novelty" in this MS, the new oxynitride SrLi₂Al₂O₂N₂:Eu²⁺ is developed from the well-known nitride narrow-band phosphors Sr[LiAl₃N₄]:Eu²⁺ by incorporating oxygen in it and the crystal structure is solved by X-ray single crystal refinement. So, 1) the narrow band of red emission is not one of the novelties in this MS. 2) Although the good thermal stability is demonstrated by Fig.4 (d), its chemical stability is bad due to its hydrolytic property. So the stability is also not one of the novelties in this MS. 3) The new material SrLi₂Al₂O₂N₂:Eu²⁺ comes from its mother system of Sr[LiAl₃N₄]:Eu²⁺ by incorporating oxygen, so the material system is not thorough one of the novelties in this MS. The author said that the structure of the ordered O-N is the key novelty. I agree with it. But the author have to demonstrate its "high performance" in advance, which make it sense for the new structure.

Answer: As the reviewer correctly pointed out, the key novelty of SALON is the O-N ordering in the structure and neither the thermal nor the chemical stability of SALON is a novelty.

The reviewer stated that the narrow band emission is not a novelty as SLA already exhibits a similar emission. While this statement is true with respect to the fwhm, the key difference between the two emissions is their spectral position. SALON is the only applicable phosphor combining an emission maximum in this spectral region with such a narrow fwhm thereby enabling a significantly higher LER (16% compared to SLA). The only other narrow band phosphor in this spectral range is SrMg₃SiN₄:Eu²⁺, which is not applicable due to its high thermal quenching. Therefore, we consider the overall emission of SALON as a novelty despite the fact that SLA already exhibits a similarly narrow emission band.

Regarding your claim that the material system is not one of SALONs novelties, we want to draw your attention to the fact that SALON is, to the best of our knowledge, the first stoichiometric compound in the Sr-Li-Al-O-N-System.

With regards to the high performance of SALON as a red phosphor, we kindly refer you to the LER gain demonstrated by the prototype and the fact that SALON meets one of the targets specified for future LED phosphors (see reply to your comment 1)).

3. How to transform from the space group of P1 (no. 2) for the matrix materials of Sr[LiAl₃N₄] system [P1 (no. 2) with a = 5.86631(12)Å, b = 7.51099(15)Å, c = 9.96545(17)Å , refer to Philipp Pust et al., Nature Materials 22 (2014) 1-6,] to the space group of P4₂/m (no. 84) for SrLi₂Al₂O₂N₂ by incorporation of oxygen? When does it start to transform to be the ordered O-N lattice at the critical point of the oxygen concentration? The XRD patterns of different doping concentration of oxygen should be appended. In this figures, the transitional process from disordered O-N to the order O-N lattice should be illustrated. Abrupt change or gradient change? In addition, how about the evolution of the SrO impurity with increasing oxygen concentration?

Answer: The two compounds (SALON and SLA) share UC₄C₄ as their aristo-type, yet we deliberately did not go on to explain the crystallographic details in the manuscript as we deem such discussions to be far from the readers interest since Nature Communications is not aimed at a crystallographic audience.

Concerning the transition from SLA to SALON, you are absolutely right in that this is an interesting field of study. We did investigate different oxygen concentrations and found that the transition involves a third, intermediate structure. This intermediate compound also exhibits a different luminescence behaviour and a large phase width due to the presence of mixed oxygen/nitrogen sites. We already collected a lot of data on this compound, yet we do not see any possibility of discussing all of this in the manuscript at hand. The intermediate phase will be the subject of a future publication, where we will be able to discuss it in detail.

4. Because the impurity of SrO is easily decomposed in ambient, why do not remove the impurity of SrO in advance so as to receive the more confidential raw data, such as the XRD, PL and EDS, of the pure SrLi₂Al₂O₂N₂ ?

Answer: The decomposition of SrO results in the presence of a mixture of different strontium hydroxide phases. Furthermore, the limited stability of SALON against humidity prevents us from removing these contaminations *via* a washing process, so the samples are actually more difficult to interpret after the decomposition of the SrO.

5. It is more believable to detect whether or not the superstructure exists in SrLi₂Al₂O₂N₂ by HRTEM and neutron diffraction pattern because of the low resolution of XRD for the trace phase; furthermore,

it is more reliable to refine the fine structure of the ordered O-N in SrLi₂Al₂O₂N₂ by using neutron diffraction.

Answer: During our single crystal XRD measurements we detected superstructure reflections corresponding to the proposed structure model. As the reflections are visible in the SC-XRD data, the existence of the superstructure has already been proven. Therefore, a further verification of the superstructure using neutron diffraction or HR-TEM experiments would be redundant. As the reviewer correctly pointed out, neutron diffraction could provide a more precise refinement, especially with regards to the O/N-ordering. As the X-ray data already indicates the O/N-ordering and any other distribution of the oxygen and nitrogen atoms in the crystal structure refinement led to bond-lengths with higher standard deviations and significantly higher residuals, we chose not to conduct neutron diffraction. Instead, we used bond-length/bond-strength, MAPLE, and CHARDI calculations to further investigate the oxygen/nitrogen ordering as there are several references in the literature, wherein the assignment of nitrogen and oxygen anions within the same structure is based on these techniques.

6. As an auxiliary means, the XPS and ESP can also be used for analysis of different coordination of O and N, and the proportion of O/N in SrLi₂Al₂O₂N₂ can be detected by Oxygen and Nitrogen Analyzer.

Answer: While additional analysis using XPS and ESP could further supplement the already acquired data, we do not think that such in-depth studies are suitable for the general audience this manuscript is aimed at.

7. Concentration quenching of active ion is the basic phosphor performance, I feel it should be measured and presented.

Answer: We do agree with the reviewer that concentration quenching of SALON should be studied. However, we feel that such measurements are beyond the scope of a first report and should be the subject of subsequent work.

Reviewers' comments:

Reviewer #1 (Remarks to the Author):

The few remaining issues have been adequately addressed by the authors. Consequently, in my opinion, the paper can now be accepted for publication.

Additional Comments:

Between the identification of novel phosphor compounds and the effective implementation in commercial devices lies a long road. Hence, a report on a novel phosphor composition should give the research community sufficient details to appreciate the properties of the phosphor, including its weaknesses. The manuscript in its present form gives a fair account. For instance, it is clear that both phosphor stability and issues with the stability against moisture will require a considerable research effort, in order to be circumvented.

The key properties of the phosphor have nevertheless been described, and they are substantiated in a sufficiently reliable and detailed way. As always, adding other experiments (like neutron diffraction, electron microscopy, x-ray analysis...) will probably reveal additional information, but one should also take into account that

- 1) it is not unlikely the additional experiments – as requested by reviewer 3 – will not give final answers. For instance, the presence of the SrO impurity will obscure results from measurement techniques which are not spatially resolved. This is also pointed out by the authors.
- 2) Adding specific information like the concentration quenching of the europium dopant is indeed informative information, but most likely this will not change the basic conclusions. For phosphors where there is a less 'logic' substitution (e.g. the Eu²⁺ ion substituted for a trivalent cation), I would also have insisted on adding this to the manuscript, but in the present case I don't think this is a showstopper.

Hence, my feeling is that the paper can be accepted in its present form. I would however want to mention that when authors respond to a broad reviewer comment like " (Reviewer 3) It is necessary to provide the comparative data which include the efficiency (? lm/W), thermal and chemical stability, humidity resistance, anti-temperature-oscillation, decay behaviour and the weight of the used red phosphor powder,..." on only a few points of this extensive list, then they are placing them – maybe unintentionally – in a weak position, where you could read between the lines that there are more serious issues with this phosphor.

Reviewer #3 (Remarks to the Author):

The author should provide additional solid experimental data to answer the reviewer' comments.

Dear reviewers,

in the following, we copied the reviewer comments into this document followed by our answers (in blue). We would like to thank Reviewer #1 for final statement, which suggests to accept this manuscript for publication.

Best regards,

Hubert Huppertz

Reviewers' comments:

Reviewer #1 (Remarks to the Author):

The few remaining issues have been adequately addressed by the authors. Consequently, in my opinion, the paper can now be accepted for publication.

Additional Comments:

Between the identification of novel phosphor compounds and the effective implementation in commercial devices lies a long road. Hence, a report on a novel phosphor composition should give the research community sufficient details to appreciate the properties of the phosphor, including its weaknesses. The manuscript in its present form gives a fair account. For instance, it is clear that both phosphor stability and issues with the stability against moisture will require a considerable research effort, in order to be circumvented.

The key properties of the phosphor have nevertheless been described, and they are substantiated in a sufficiently reliable and detailed way. As always, adding other experiments (like neutron diffraction, electron microscopy, x-ray analysis...) will probably reveal additional information, but one should also take into account that

1) it is not unlikely the additional experiments – as requested by reviewer 3 – will not give final answers. For instance, the presence of the SrO impurity will obscure results from measurement techniques which are not spatially resolved. This is also pointed out by the authors.

2) Adding specific information like the concentration quenching of the europium dopant is indeed informative information, but most likely this will not change the basic conclusions. For phosphors where there is a less 'logic' substitution (e.g. the Eu²⁺ ion substituted for a trivalent cation), I would also have insisted on adding this to the manuscript, but in the present case I don't think this is a showstopper.

Hence, my feeling is that the paper can be accepted in its present form. I would however want to mention that when authors respond to a broad reviewer comment like “ (Reviewer 3) It is necessary to provide the comparative data which include the efficiency (? Im/W), thermal and chemical stability, humidity resistance, anti-temperature-oscillation, decay behaviour and the weight of the used red phosphor powder,...” on only a few points of this extensive list, then they are placing them – maybe unintentionally – in a weak position, where you could read between the lines that there are more serious issues with this phosphor.

We would like to thank Reviewer #1 for his support and his final statement that in his eyes the paper can now be accepted for publication. Concerning the comments of Reviewer #3, we discussed all points in our last response to the reviewers and provided the reasons.

Reviewer #3 (Remarks to the Author):

The author should provide additional solid experimental data to answer the reviewer' comments.

Reviewer #3 demands additional experimental data. As we stated in our answer to his suggestions, these measurements are not necessary for the conclusions drawn in this manuscript. Additionally, as Reviewer 1 correctly pointed out, the results would probably not be more reliable than the data already at hand.

We did address any specific concerns that were raised regarding the claims in our manuscript, such as the N/O-ordering, for which we included the additional MAPLE calculations. Therefore, we do not see how the addition of even more experimental data could reinforce the findings we aim to present.